# Novel Bio-Based Materials: From Castor Oil to Epoxy Resins for Engineering Applications

**DOI:** 10.3390/ma16165649

**Published:** 2023-08-16

**Authors:** Constantin Gaina, Oana Ursache, Viorica Gaina, Alexandru-Mihail Serban, Mihai Asandulesa

**Affiliations:** “Petru Poni” Institute of Macromolecular Chemistry, 41A Gr. Ghica Voda Alley, 700487 Iasi, Romania; gcost@icmpp.ro (C.G.); oana.buliga@icmpp.ro (O.U.); serban.alexandru@icmpp.ro (A.-M.S.); asandulesa.mihai@icmpp.ro (M.A.)

**Keywords:** vegetable oil, castor oil, epoxy resin, thermal characterization

## Abstract

The paper presents the synthesis and thermal behavior of novel epoxy resins prepared from epoxidized castor oil in the presence of or without trimethylolpropane triglycidyl ether (TMP) crosslinked with 3-hexahydro-4-methylphtalic anhydride (MHHPA) and their comparison with a petroleum-based epoxy resin (MHHPA and TMP). Epoxidized castor oil (ECO) was obtained via in situ epoxidation of castor oil with peroxyacetic acid. The chemical structures of castor oil (CO), ECO, and epoxy matrix were confirmed using FT-IR and ^1^H-NMR spectroscopy. The morphological and thermal behavior of the resulting products have been investigated. Compared to petroleum-based resins, castor oil-based ones have a lower *T_g_*. Anyway, the introduction of TMP increases the *T_g_* of the resins containing ECO. The morphological behavior is not significantly influenced by using ECO or by adding TMP in the synthesis of resins. The dielectric properties of epoxy resins have been analyzed as a function of frequency (1 kHz–1 MHz) and temperature (−50 to 200 °C). The water absorption test showed that as *T_g_* increased, the percent mass of water ingress decreased.

## 1. Introduction

Nowadays, there is a great interest in replacing synthetic polymers with natural ones. Besides the intuitive reasons (biodegradability, biocompatibility, and renewable resources), they are also used due to their versatility derived from their high reactivity and various functional groups such as hydroxyl, carboxylic acid, and so on. These polymers are either of animal or vegetable origin, and they are proteins or polysaccharides such as gelatin, collagen, cellulose, starch, chitosan, alginate, pullulan, and many others [1,2]. Additionally, considerable research was directed toward the development of polymers from triglyceride oils as a natural alternative to petroleum-based ones.

The technical applications of industrial vegetable oils are linked to their physicochemical characteristics, the chain length, the degree or nature of unsaturation, or the presence of special functional groups attached to the fatty acid chain [3,4,5]. Vegetable oils are generally triglycerides of various fatty acids with varying degrees of unsaturation and are used in both the food and non-food industries, such as the production of coatings, lubricants, adhesives, molding compositions, and paints [6,7,8,9]. Over time, remarkable progress was registered in the use of these vegetable oils to obtain polymers that present an impressive range of structures, properties, and promising applications [2,10,11,12]. Norbornylized and alkyd vegetable oils were obtained and further used in different formulations to obtain thermosets with applications mainly in the field of composites or coatings [13,14,15,16].

Biobased epoxies (epoxidized vegetable oils (EVOs)) have gained great interest in the chemical industry because they can successfully replace petroleum-based epoxies and can be obtained from vegetable oils that have a low cost, are non-toxic and biodegradable and can be used as reactive modifiers and diluents for epoxy composites. Epoxidized vegetable oils may be used either non-modified or functionalized via epoxy ring opening with various compounds: amines, anhydrides, allylic alcohols, thiols, and α, β-unsaturated acids [17,18,19,20,21,22,23,24]. Thus, EVO was crosslinked with anhydrides and then used as obtained or combined with biofillers: keratin, lignin, spruce bark powder, hydrochar, lignocellulosic biomass, and flax woven fibers [25,26,27,28]. Carboxylic acids were also used as hardeners in the process of obtaining epoxidized thermosets based on vegetable oils [29,30,31,32,33]. EVO-based vitrimers have been synthesized via three chemistries, namely, transesterification, Schiff base, and disulfide exchange [34]. The use of vegetable oils as alternatives to petroleum-based materials for polyurethane production was another point of interest for scientific researchers [35,36].

The present paper deals with a comparative study of MHHPA-cured epoxidized castor oil (ECO) with petro-based TMP epoxy. The molecular structure of the resulting compounds was identified with FT-IR and ^1^H-NMR. The obtained resins were characterized in terms of thermal decomposition, morphology, dielectric properties, and solubility. The advantage of choosing ECO in the design of resins is its renewable source.

## 2. Materials and Methods

### 2.1. Reagents and Materials

Castor oil, 3-hexahydro-4-methylphtalic anhydride (MHHPA), trimethylolpropane triglycidyl ether (TMP), aqueous hydrogen peroxide (30 wt.%), glacial acetic acid (99.7 wt.%), tetrabutylammonium chloride (TBAC), and sulfuric acid (96 wt.%) were purchased from Sigma Aldrich (Sigma-Aldrich, Darmstadt, Germany).

### 2.2. Films Preparation

The ECO was synthesized according to the method described in the literature [37].

To design the epoxy resins (P1-P4), the 1:1 stoichiometric ratio of the epoxy groups to the anhydride groups was used (Figure 1).

The proper amount of ECO or TMP was heated at 60 °C to decrease its viscosity. Then, the required amount of MHHPA was added to the ECO or ECO-TMP mixture that was subjected to ultrasound for 30 s. To facilitate the reaction, TBAC was used as the initiator, being added in a proportion of 1 wt% based on the total weight of the curing agent-epoxy mixture. The homogenous blends were poured into silicon molds and introduced into a preheated convection oven for curing and post-curing treatment. For complete curing of the resins, the silicon molds were heated at 140 °C for 3 h and 180 °C for 2 h. The material designation and composition of the ECO-based products are summarized in Table 1.

## 3. Measurements

The infrared spectra of the samples were obtained in Attenuated Total Reflectance (ATR) geometry by using a Bruker Vertex 70 spectrometer (Ettlingen, Germany) equipped with a Specac™ ATR unit. The sample surface was scanned with a resolution of 2 cm^−1^ in the 4000 cm^−1^–600 cm^−1^ range of wavenumber.

The proton nuclear magnetic resonance spectra were recorded on a Bruker NMR spectrometer, the Avance DRX 400MHz (Rheinstteten, Germany), using CDCl_3_ as solvent and tetramethylsilane as an internal standard.

The thermal characterization of epoxy resins was studied using a Jupiter STA 449 F1 thermobalance (Netzsch, Selb, Germany) coupled online with an FT-IR spectrometer Vertex 70 (Brüker, Germany) and a mass spectrometer Aëolos QMS 403C (Netzsch, Germany). Here, a 15 mg sample was placed in Al_2_O_3_ crucibles and heated in dynamic mode with a 10 °C/min heating rate. Runs were made in a nitrogen atmosphere with a 50 mL/min flow from room temperature up to 675 °C. The gases that resulted during the thermal degradation processes were transferred to the TGA-IR external modulus, equipped with an MCT (Mercury Cadmium Telluride) detector, through a Teflon line maintained at 190 °C, and to the mass spectrometer through a quartz capillary of 75 μm diameter maintained at 290 °C. FT-IR spectra were registered in 3D size and analyzed with OPUS 6.5 software. Mass spectrometer working parameters were ionization energy with an electron impact of 70 eV and vacuum of 10^−5^ mbar. MS spectra were acquired with Aëolos 32 software, up to 200 *m*/*z*.

Differential scanning calorimetry (DSC) measurements were conducted on a DSC 200 F3 Maia (Netzsch, Germany). About 9 mg of sample were heated in pressed and punched aluminum crucibles at a heating rate of 10 °C/min. Nitrogen was used as an inert atmosphere at a flow rate of 100 mL/min.

The dielectric measurements were carried out with the Novocontrol Concept 40 Broadband Dielectric Spectrometer (Novocontrol Technologies, Montabaur, Germany). The samples were placed between two gold-plated flat electrodes, and the system was introduced into the cryostat of the device. The alternating electrical field oscillations were provided by an Alpha-A High-Performance Frequency Analyzer in a frequency window between 10^0^ and 10^6^ Hz. The temperature was applied between −50 °C and 200 °C and assisted by a Quatro Cryosystem device.

Scanning electron microscopy (SEM) analysis of fractures and surfaces of the polymeric films was performed with a Vega Tescan microscope after sputter-coating the specimens with gold. The surface morphology was investigated with a SOLVER PRO-M atomic force microscope.

A water absorption test was conducted based on the ASTM D570 standard, in which the samples were soaked in boiling water (at 100 °C) and water at room temperature for 24 h to establish the relationship between the effect of ECO loading in the epoxy system and the water absorption capability of the system. Firstly, samples (60 × 60 × 1 mm^3^) were dried in an oven at 50 °C for 24 h, cooled in a desiccator, and then weighed (*W*_0_) with a precision balance. For the 24-h immersion procedure (ASTM D570 standard), the conditioned specimens were entirely immersed in distilled water at room temperature or in boiling water and maintained for 24 h. At the end of the 24 h, the tested samples were removed from the water, carefully wiped with filter paper, and their mass was measured (*W_t_*). The moisture content ratio (*WA*, %) was calculated using Equation (1) [38].
(1)$$WA,\%=\frac{W_{t}-W_{0}}{W_{0}}\times 100$$
where *W*_0_ represents the conditioned weight of the tested sample, and *W_t_* is the wet mass of the tested sample after 24 h of immersion.

## 4. Results

### 4.1. FT-IR and ^1^H-NMR Characterization

In this study, the polyester resins were obtained by the reaction of epoxidized castor oil with MHHPA and TMPTGE. The curing of epoxy resins with cyclic anhydrides is started by the reaction of anhydride groups with hydroxyl groups that are present in the reaction mixture (as part of the epoxy resin) to generate a monoester and a carboxylic acid group. The carboxylic acid group then reacts with the epoxide to form a diester with a hydroxyl group, which in turn reacts with another anhydride [39]. The chemical structure of the obtained resins was identified by FT-IR and ^1^H-NMR spectroscopy. Figure 1 exhibits the FT-IR spectra of CO, ECO, and P(1–4). The FT-IR spectrum of CO (Figure 1a) shows the absorption bands at around 3400 cm^−1^ specific to the O-H stretching vibration, at 2928 cm^−1^ (-CH_2_, asymmetric stretching) and 2855 cm^−1^ (-CH_2_ symmetric stretching), at around 3010 cm^−1^, 1655 cm^−1^, 858 cm^−1^ and 725 cm^−1^ characteristic to the carbon-carbon double bond of castor oil, and at 1745 cm^−1^ due to the C=O group [40,41]. In the FT-IR spectrum of ECO, the absorption band specific to the C=C bonds at 3010 cm^−1^ disappeared (which indicates the complete conversion of double bonds in the oxirane ring) and a new peak appeared at 843 cm^−1^ due to the formation of an oxirane ring. In the case of polymers (Figure 1b), the disappearance of the absorption band characteristic of the oxirane ring can be observed (except for P2, where a small amount of oxirane ring is still present). Other absorption bands appeared in the FT-IR spectra of all of these samples were, at 3500–3360 cm^−1^ (O-H stretching vibration), at 2929 cm^−1^ (-CH_2_, asymmetric stretching), and 2857 cm^−1^ (-CH_2_ symmetric stretching), at 1740 cm^−1^ (triglyceride carbonyl stretching), at 1461 cm^−1^ (CH_2_ bending vibration), a peak at 1375 cm^−1^ (CH_3_ symmetrical bending vibration), and peaks at 1240, 1158, and 1100 cm^−1^ due to the stretching vibrations of the C−O group in esters [42].

Figure 2 shows the ^1^H-NMR spectra of CO and ECO. Compared with the ^1^H-NMR spectrum of CO, where the proton signals in the 5.42–5.58 ppm range associated with carbon-carbon double bond bands disappeared the signal at 3.39 ppm present in the ECO spectrum ascribed to epoxy groups demonstrates that the unsaturation C=C presented in CO was successfully replaced with the epoxy group of ECO [37,43,44].

### 4.2. Thermal Analyses

The TG and DTG curves obtained for epoxy polymeric network samples are illustrated in Figure 3. In addition, Table 2 presents the main thermal parameters extracted from the TG and DTG curves: *T_onset_*—temperature at which the thermal degradation stage begins; *T_peak_*—temperature where the thermal degradation rate is maximum; W—weight loss at the end of each stage; residual mass at 675 °C; *T*_5%,_ *T*_10%_, and *T*_30%_ temperatures corresponding to 5%, 10%, and 30% weight loss, respectively; *T_S_*—statistic heat resistance; *T_GS_*—temperature assigned to the maximum quantity of gases released (peaks of the Gram–Schmidt curve). The statistic heat-resistant index (*T_S_*) was calculated to evaluate the thermal stability of the sample. This index corresponds to the temperature of the polymer within the physical heat resistance tolerance limit [45,46] and can be calculated with Equation (2), as follows:*Ts* = 0.49[*T_5%_* + 0.6(*T_30%_* − *T_5%_*)] (2)

The thermal degradation of epoxy polymeric networks in the absence of oxygen is a complex process that takes place in several stages as a function of the sample composition and the ratio between the constituents. As DTG curves show, P1, P3, and P4 present a three-stage decomposition process, while P2 presents only a one-stage process characterized by a sharp, narrow peak. This is due to the presence of epoxidized castor oil, a biobased vegetable oil derivate with hydroxyl groups in its structure, in different ratios for P1, P3, and P4 in comparison with P2, which is a petroleum-based resin.

The P1 sample, which contains epoxidized castor oil crosslinked with 3-hexahydro-4-methylphtalic anhydride, exhibits a three-stage consecutive degradation process with a *T_onset_* at 228 °C, a maximum degradation rate at 366 °C, and a residual mass value of 3.16%. This type of behavior is specific to samples that contain vegetable oils as major constituents at lower temperatures, due to higher instability, oxygenated functional groups are subjected to decomposition reactions, while at higher temperatures, the aliphatic chains are cracked apart through C–C bond breaking [47,48,49]. Additions of trimethylol-propane triglycidyl ether in P3 and P4 in various ratios led to a thermal decomposition process with three stages but with one major, sharp peak and two minor shoulder peaks, one before and one after the major peak. Independent of the ratio between constituents, the degradation process begins nearly simultaneously for P3 and P4, at 214 and 210 °C, respectively, and registers the maximum degradation rate at almost similar temperatures, 393 and 392 °C, respectively. One possible explanation is that both samples undergo the same decomposition reaction. However, P4 has a smaller residual mass, mostly because of the ratio between constituents: a higher content of epoxidized castor oil and a smaller content of 3-hexahydro-4-methylphtalic anhydride and trimethylol-propane triglycidyl ether.

On the other hand, P2, a petroleum-based epoxy polymeric network obtained through the crosslinking reaction of TMP with MHHPA, exhibits only one decomposition stage, which begins at 355 °C and has a sharp DTG peak with a maximum at 387 °C. Taking into account *T_s_* and T_10_ parameter values (Table 2), the following thermal stability series can be established: P1 < P3 < P4 < P2. The most thermally stable polymeric network is P2 due to its higher reticulation degree and dense structure with a small free space volume.

#### 4.2.1. Evolved Gas Analysis

Thermogravimetric analysis coupled online with FT-IR and MS spectroscopy enables the identification of the main gases produced during the thermal degradation of the samples in a nitrogen-inert atmosphere. Due to the similar composition of P3 and P4, only P1, P3, and P2 samples were chosen for evolved gas analysis. Figure 4 represents the 3D spectrum of the evolved gases continuously recorded from room temperature up to 675 °C. At the temperatures at which the maximum amount of gas was recorded (Gram–Schmidt curve peaks near the DTG peaks), 2D FT-IR spectra were extracted (Figure 5).

Thermal degradation of the polymeric networks based on vegetable oils generally begins with the breaking of the carboxylic functional groups into hydroxyls and esters. With the increasing temperature, the C-C bond scission as well as the dehydrogenation reactions occur and lead to the formation of aliphatic and cyclic hydrocarbons, carbon dioxide, carbon monoxide, and carbonylic species like aldehydes and ketones [47,48,49]. As can be seen from the 2D FT-IR spectra, the main volatile products that could be identified for the three samples are similar, regardless of the temperature. The most noticeable difference between spectra occurs in terms of the intensities of the bands. The most significant bands identified in the infrared spectra are located between 3900 and 3400 cm^−1^ being assigned to stretching vibration of the free and intramolecular bound hydroxyl groups, 3000 and 2800 cm^−1^, region of asymmetric and symmetric stretching of CH_3_, CH_2,_ and CH_,_ 2450–2100 cm^−1^ specific to stretching vibration of O=C=O, 1880–1650 cm^−1^, stretching vibration of C=O bonds, 1600–1400 cm^−1^, bending vibration of C-H, 1270–1000 cm^−1^ characteristic to stretching vibration of C-O and C-O-C bonds, 1000–800 cm^−1^ corresponding to bending vibration of unsaturated aliphatic fragments [45,48,49,50]. All spectra present the TGA-IR external module MCT detector-specific wide “ice band”, recorded at about 3250 cm^−1^. Based on the identified bands, the most abundant volatile compounds are: carbon dioxide (2350 and 670 cm^−1^), water and alcohols (3870, 3740, 3580, and 1390 cm^−1^), carbon monoxides (2180 cm^−1^), aldehydes, ketones, and esters (1878, 1805, 1720 and 1118 cm^−1^), aliphatic saturated and unsaturated hydrocarbons (2953, 2877, 1635, 995 and 910 cm^−1^). Increased intensities of OH, C=O, and C-O specific bands presented in FT-IR spectra at the beginning of the degradation process (220–350 °C) indicate that alcohol and esters are primarily released. As the temperature increases, spectral area intensities specific to oxygen-containing products decrease gradually while the intensities of bands assigned to aliphatic or cyclic saturated and unsaturated hydrocarbons remain constant, showing that this type of compound is released mostly at higher temperatures.

The mass spectrometry confirms the volatile compounds identified through FT-IR spectroscopy. The assignments of ionic fragments identified from *m*/*z* signals extracted at the temperature of the maximum amount of volatile product release (*T_GS_*) (Figure 6) were done in accordance with NIST spectral libraries and the literature [45,48,50,51]. Table 3 presents the most important volatile compounds identified for epoxy polymeric networks.

During the degradation process of epoxy polymeric networks, numerous volatile compounds are released. As Table 3 shows, the major compounds identified during the process were: alcohols (methanol, ethanol, and propanol), aldehydes, and ketones (formaldehyde, acetone, octan 2-one, heptanal), esters and ethoxy (butanoic acid, 1-methylbutyl ester, diethoxy methane), saturated or unsaturated aliphatic and cyclic hydrocarbons (methane, propane, butane, cyclohexane, methylcyclohexane, propene, hexadiene, cyclohexene, 3-methylcyclohexene, 1,4-cyclohexadiene). All the obtained data are in good agreement with those presented in the literature [45,48,50,51].

#### 4.2.2. DSC Measurements

Generally, the *T_g_* values are strongly dependent not only on the EVO structure but also on the nature and structure of the hardener. The *T_g_*s of P(1–4) were determined by DSC, and the results are presented in Figure 7 and Table 2.

As can be seen from the figure, sample P1, based on ECO and MHHPA, has the lowest *T_g_* (6.83 °C), due to the large volume of ECO. On the other hand, the sample based on MHHPA and TMP (P2) has a higher *T_g_* (58.65 °C), which could be related to an increased crosslinking density. The introduction of TMP in the ECO/MHHPA system leads to an increase in *T_g_*, and as the amount of TMP increases, so does *T_g_*, but not higher than that of the sample based on MHHPA and TMP (P2).

### 4.3. SEM

The cross-sectional morphological structure of the experimental films was observed by SEM, as shown in Figure 8. The samples were clear and transparent, and macroscopic phase separation was not observed.

### 4.4. Dielectric Measurements

Figure 9 shows the evolution of the dielectric constant (ε’) with frequency (a) and temperature (b) for all samples investigated here. The isothermal plots retrieved in Figure 9a reveal a progressive decrease in ε’ towards increasing frequency due to the ability of chemical dipoles to follow the oscillations of the external alternating electrical field [52].

The COE product cross-linked with the MHHPA agent exhibits the highest magnitude of ε’ for the entire frequency window. During the crosslinking process, some chemical dipoles disappear, such as anhydride groups from the MHHPA agent, and other chemical dipoles are formed, such as hydroxyl and ester groups [53]. The newly produced dipolar units may account for the dipolar activity of the P1 compound. On the other hand, the dielectric constant of the P2 sample is lower than that of the P1 sample because the MHHPA product cross-linked with the TMP agent leads to a material with lower polarizable units than that of the MHHPA-based material. As a consequence, the P2 sample polarizes less in the presence of the electrical field. Furthermore, we notice that the dielectric constant of the P3 and P4 samples is considerably lower than that of the P1 and P2 systems. The mixture of the crosslinking agents disturbs the molecular conformation of the polymer by increasing the free volume. As a consequence, the polarization of the pair of COE units cross-linked with MHHPA/TMP mixtures is lowered due to the reduced number of chemical dipoles per volume unit. The numerical values of the dielectric constant retrieved at various frequencies are listed in Table 4.

The isochronal plots in Figure 9b show a big dependence of the dielectric constant on temperature. Hence, the dielectric spectra may be divided into two different temperature regions: a low-temperature region, where ε’ increases slowly with temperature (e.g., for P4 system, at −50 °C, ε’ = 4.1, while at 25 °C, ε’ = 4.5), and high-temperature region, where ε’ increases markedly with temperature (e.g., for P4 system, at 200 °C, ε’ = 673.7). The transition between the low- and high-temperature regions coincides with the appearance of the glass transition temperature of the systems.

The dielectric loss (ε”) is related to the energy dissipated by the chemical dipoles to follow the oscillations of the electrical field. The evolution of ε” with frequency for all samples is presented in Figure 10a. The high losses observed at low frequencies for P1 and P3 systems are related to the dipolar α-relaxation process, which is connected to the glass temperature of samples. At high frequencies, the latter signal is diminished. With further increase in frequency, the dielectric loss of the samples is low, <0.16 (f = 10^4^ Hz). According to Figure 10b, at low temperatures, the magnitude of ε” is reduced and related to the dipolar relaxation processes of the materials. On the other hand, at high temperatures, the mobility of charge carriers is activated and increases substantially the amplitude of ε”.

The dielectric spectra of the electrical modulus (M”) enclose the signals related to the dipolar relaxation phenomena as well as the transport of charge carriers. The frequency dependences of M” selected at room temperature (Figure 11a) reveal the characteristic signal of dipolar α-relaxation. The latter is detected at low frequencies as a high signal for P1 and P3 samples and a lower signal for P4 samples. Following Figure 11b, a dielectric peak related to the conductivity of charge carriers is detected above 50 °C for all investigated samples.

The evolution of measured conductivity (σ) as a function of frequency is presented in Figure 12 at room temperature (a) and at 60 °C (b) for all investigated samples. At room temperature (Figure 12a), σ increases progressively towards increasing frequency. This regime is generally assigned to the dipolar relaxation-type phenomena of the materials [54]. The low values of conductivity detected at low frequencies (e.g., at 1 Hz, σ values are recorded between 8 × 10^−13^ S/cm and 3.6 × 10^−14^ S/cm) reveal the insulating character of the investigated systems. At 60 °C (Figure 12b), however, the σ(f) profiles reveal a flat plateau region at low frequencies. The regime where the measured conductivity is independent of frequency is generally characteristic of the free charge carriers moving through the material lattice [54]. The frequency-independent plateau is observed for P1 and P3 systems.

### 4.5. Water Absorption

Water absorption in epoxy systems is relevant because this phenomenon will tend to deteriorate the properties of the system. As can be seen from Table 4, the water absorption at room temperature of epoxy resins based on ECO decreases by adding TMP (from 9.41 to 1.74% for P3 and 2.60% for P4) but is higher than that of P2 due to the presence of hydroxyl groups in the oil structure. Additionally, Table 2 and Table 4 indicate that as the *T_g_* increases the percent mass of water ingress decreases. On the other hand, the water absorption at 100 °C of all samples is approximately the same and is located in the range of 3.06–3.68%. The relatively high water absorption values may be related to the experimental temperature, which could accelerate the diffusion of H_2_O molecules into the polymer.

## 5. Conclusions

The new epoxy resins based on ECO were characterized by FT-IR and ^1^H-NMR spectroscopy for structural confirmation. The thermal stability and *T_g_* of the ECO-based resins are lower than those without ECO. *T_g_* varied in the temperature region of 6.83–58.65 °C, with the ECO-MHHPA-TMP resins having a *T_g_* between that of the ECO-MHHPA resin and that of the MHHPA-TMP resin. For the comparative epoxy resins, no microscopic phase separation was observed at any loading level. The main volatile products from thermal degradation of all epoxy resins, identified by TG-MS-FTIR analysis, revealed alcohols, aldehydes and ketones, esters and ethoxy, saturated or unsaturated aliphatic and cyclic hydrocarbons.

Compared with P2 epoxy resins, the ECO-MHHPA-TMP epoxy resins exhibited lower dielectric constants and higher water absorptions, which were attributed to the ECO and incorporation of TMP in the structure of the epoxy resins.

## Data Availability

Not applicable.

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
