# Peer review of "Novel Bio-Based Materials: From Castor Oil to Epoxy Resins for Engineering Applications"

_materials, 2023, doi:10.3390/ma16165649_

Round 1

Reviewer 1 Report

This manuscript describes the synthesis and thermal behavior of epoxy resins prepared from epoxidized castor oil. The as-synthesized resin was characterized by FTIR, NMR, and thermal analyses. The paper is flawless, and worthy of publication. However, the following minor changes need to be done before publication:

1. All the figures are too small to be seen properly, especially NMR diagram is mostly unreadable. Please enlarge.

2. It would be better if authors incorporate two tables, each for FTIR and NMR, to explain the attributions of peaks and point out the changes.  

Author Response

1. Figures 1-8 have been enlarged.

2. We improved figures 1a and 1b with the corresponding signals and additional comments was inserted in the text.

Reviewer 2 Report

The study of Novel bio-based materials: from castor oil to epoxy resins for engineering applications is indeed very relevant research keeping in mind the demand for sustainable polymer products.

1.     There here are some typos (typing errors) throughout that will need to be revised. Nothing major, so a quick run-through of the manuscript and corrections should do the work. Example: Line number 122, H-NMR, the conclusion is the subheading “4”. Not 5.

2.      Introduction starts very little to none about the various research work significance to vegetable oil epoxides, or castor oils in general. It would also be beneficial to add the various recent literature that talks about epoxidized vegetable oil-based thermosets and composites with natural fillers, glass fiber reinforcement etc. most common example of labs working exclusively into epoxy vegetable oils, natural fillers, fibers, thermoset and composite systems would be recent papers from notably, Alice Mija, Mark D Soucek, Chiara Di Mauro, Jomin thomas, Ica Manas-Zloczower among various others. Review articles are also available which is of huge importance to vegetable oil epoxides. Castor oil as such is also used in acrylic polymers for polyurethane coatings and adhesives. To make authors understand that the end applications of biobased epoxides whose study is conducted here expands the scope and opportunity of the discussed research.

3.     Please include Scan details in FTIR and scanning temperature range for equipment like DSC. Most of the figure qualities (2,4,5,6) need to be improved though the data is sound.

Overall it’s a very well-written manuscript giving information on a significant study on vegetable oil epoxides. 

Overall it’s a very well-written manuscript giving information on a significant study on vegetable oil epoxides. Needs minor revision on the points mentioned. The introduction is near to none without presenting the importance and significance of the work and how it compares to the existing literature on vegetable oil epoxides. 

Author Response

1. We revised and made the corrections in the manuscript.

2. We modified the introduction part, taking into account the suggestions of the referent.

3. We entered the required information in the text and improved figures 1-8.

Reviewer 3 Report

Dear Authors, the field of bio-based materials obtained from renewable sources is especially interesting from an environmental point of view. The authors produced a resin with ECO and compared it with petroleum-based ones. Tg values obtained ​​are lower, after that Authors added TMP and Tg increased, but TMP is a synthetic derivative. The scope of the paper is not clear. I suggest these recommendations before the paper is evaluated further.

1)    Introduction is too concise and lacking in content.

2)    Introduction. Lines 23-25.

References are necessary when Authors describes general statements.

3)    Introduction. Lines 28-31.

I suggest to add a discussion of common polymers derived from petroleum sources, examples of bio-based polymers erived from renewable sources and the relative literature comparison, specifying any improvements. More discussion on these part and more references are necessary. I suggest these two recent references on the topic.

-          Indranil Chakraborty and Kaushik Chatterjee, Biomacromolecules 2020 21 (12), 4639-4662

DOI: 10.1021/acs.biomac.0c01291.

-          Maiuolo, L.; Olivito, F.; Algieri, V.; Costanzo, P.; Jiritano, A.; Tallarida, M.A.; Tursi, A.; Sposato, C.; Feo, A.; De Nino, A. Synthesis, Characterization and Mechanical Properties of Novel Bio-Based Polyurethane Foams Using Cellulose-Derived Polyol for Chain Extension and Cellulose Citrate as a Thickener Additive. Polymers 2021, 13, 2802. https://doi.org/10.3390/polym13162802

4)    Authors should describe what thermosets are and relative applications.

5)    Scheme 1 .

What is “COE”?

Labels are not clear.

6)    Scheme 1. Is this synthesis novel or not ? Authors should insert a scheme about reaction mechanism. Which functional groups react? How does react the functional groups and where?

7)    Figure 1. Authors should add the functional groups on FT-IR scheme for each signal.

8)    During FT-IR discussion signals-functional groups references are necessary.

9)    Figure 2. The resolution of the structure is really poor. Authors should increase the size.

10)                        Sub-paragraph 3.6 is water absorption, not solubility.

11)                        What is the advantage of having produced the resin with eco? Is there any advantage?

Minor English editing required

Author Response

1-4. The reviewer`s comments are justified and we took them into consideration. The subject is well known to experts in the field, as indicated by the numerous reports in the literature, and briefly presented herein. This is a research paper, so the introduction should not exceed a certain extent. Even so, we have added supplemental information as the reviewer suggested.

5. We changed the scheme, instead of COE it is ECO.

6. The scheme has been changed. We have included in the text a comment regarding the reaction mechanism studied by other researchers.

7. We modified the figure.

8. We have included references in the text.

9. The resolution of Figure 2 was increased.

10. We corrected the text.

11. The advantage is that a part of the raw materials is from renewable resources.

Round 2

Reviewer 3 Report

Dear Authors, before the article will be accepted, you have to follow the comments and suggestion of point 3 and 11. 

10)

  Introduction. Lines 28-31.

I suggest to add a discussion of common polymers derived from petroleum sources, examples of bio-based polymers erived from renewable sources and the relative literature comparison, specifying any improvements. More discussion on these part and more references are necessary. I suggest these two recent references on the topic.

-          Indranil Chakraborty and Kaushik Chatterjee, Biomacromolecules 2020 21 (12), 4639-4662

DOI: 10.1021/acs.biomac.0c01291.

-          Maiuolo, L.; Olivito, F.; Algieri, V.; Costanzo, P.; Jiritano, A.; Tallarida, M.A.; Tursi, A.; Sposato, C.; Feo, A.; De Nino, A. Synthesis, Characterization and Mechanical Properties of Novel Bio-Based Polyurethane Foams Using Cellulose-Derived Polyol for Chain Extension and Cellulose Citrate as a Thickener Additive. Polymers 2021, 13, 2802. https://doi.org/10.3390/polym13162802

10) 

What is the advantage of having produced the resin with eco? Is there any advantage?

Authors must report a scientific explanation in the manuscript

Minor English editing required

Author Response

3 -  We added in the first part of the introduction a short discussion about bio-based polymers derived from renewable sources as the reviewer suggested and we added the suggested literature.

11 - We reported in the manuscript (at the final of introduction section) what advantage we thought when choosing ECO.